# Value of the Lung Immune Prognostic Index in Patients with Non-Small Cell Lung Cancer Initiating First-Line Atezolizumab Combination Therapy: Subgroup Analysis of the IMPOWER150 Trial

**DOI:** 10.3390/cancers13051176

**Published:** 2021-03-09

**Authors:** Ashley M. Hopkins, Ganessan Kichenadasse, Ahmad Y. Abuhelwa, Ross A. McKinnon, Andrew Rowland, Michael J. Sorich

**Affiliations:** 1College of Medicine and Public Health, Flinders University, Bedford Park, SA 5042, Australia; ganessan.kichenadasse@flinders.edu.au (G.K.); ahmad.abuhelwa@flinders.edu.au (A.Y.A.); ross.mckinnon@flinders.edu.au (R.A.M.); andrew.rowland@flinders.edu.au (A.R.); michael.sorich@flinders.edu.au (M.J.S.); 2Department of Medical Oncology, Flinders Centre for Innovation in Cancer, Flinders Medical Centre, Bedford Park, SA 5042, Australia

**Keywords:** atezolizumab, non-small cell lung cancer, overall survival, lung immune prognostic index

## Abstract

**Simple Summary:**

The lung immune prognostic index (LIPI) is proposed as a simple risk scoring tool to differentiate differences in survival from immune checkpoint inhibitors (ICIs) in advanced non-small cell lung cancer (NSCLC). The tool has not been evaluated for performance in a NSCLC cohort initiating first-line, combination ICI approaches. In a large cohort of participants with chemotherapy-naïve, metastatic non-squamous NSCLC, we independently validated, for the first time, that LIPI discriminates a clear subgroup of patients likely to achieve reduced survival following the initiation of combination therapies including the ICI atezolizumab.

**Abstract:**

The lung immune prognostic index (LIPI) is proposed to differentiate prognosis and treatment benefit from immune checkpoint inhibitors (ICIs) in advanced non-small cell lung cancer (NSCLC). There is minimal information on the predictive importance with first-line, combination ICI approaches. In post-hoc analysis of IMpower150, Cox-proportional hazard analysis assessed the association between LIPI groups and overall survival (OS)/progression free survival (PFS). IMpower150 involved chemotherapy-naïve, metastatic non-squamous NSCLC participants randomized atezolizumab-carboplatin-paclitaxel (ACP), bevacizumab-carboplatin-paclitaxel (BCP), or atezolizumab-BCP (ABCP). Good (0 factors), intermediate (1 factor), and poor LIPI (2 factors) were defined via derived neutrophil-to-lymphocyte ratio >3, and lactate dehydrogenase >upper limit of normal. Of 1148 participants, 548 had good, 479 intermediate, and 121 poor LIPI. In 385 participants randomised ABCP, a significant association between LIPI and OS (HR (95%CI): intermediate LIPI = 2.16 (1.47–3.18), poor LIPI = 5.28 (3.20–8.69), *p* < 0.001) and PFS (HR (95%CI): intermediate LIPI = 1.47 (1.11–1.95), poor LIPI = 3.02 (2.03–4.50), *p* < 0.001) was identified. Median OS was 24, 16, and 7 months for good, intermediate, and poor LIPI, respectively. ACP associations were similar. Relative OS treatment effect (HR 95%CI) of ABCP vs. BCP was 0.78 (0.53–1.15), 0.67 (0.49–0.91), and 0.87 (0.51–1.47) for the good, intermediate, and poor LIPI groups, respectively (P(interaction) = 0.66), with no benefit in median OS observed in the poor LIPI group. LIPI identified subgroups with significantly different survival following ABCP and ACP initiation for chemotherapy-naïve, metastatic non-squamous NSCLC. There was insufficient evidence that LIPI identifies patients unlikely to benefit from ABCP treatment.

## 1. Introduction

Standard first line treatment of advanced non-small cell lung cancer (NSCLC) is quickly evolving to be immune checkpoint inhibitors (ICIs) in combination with conventional chemotherapies [1]. With the arrival of ICIs, a multitude of potential factors that may assist in explaining heterogeneity in survival outcomes with immunotherapies have emerged [2,3,4,5,6,7,8,9,10]. However, for many of these factors, there is minimal information on performance with first-line, ICI combination approaches. One such factor is the lung immune prognostic index (LIPI).

LIPI is calculated based on pre-treatment derived neutrophil to lymphocyte ratio (dNLR) and lactate dehydrogenase (LDH) levels. It has been demonstrated as a prediction tool that differentiates likely prognosis of patients with NSCLC initiating ICI treatment [8,11,12,13,14,15,16]. Studies have also demonstrated that LIPI may differentiate prognosis in patients treated with ICIs in other cancer types, including breast cancer, hepatocellular carcinoma, melanoma, renal cell carcinoma, small cell lung cancer, and urothelial carcinoma [12,17,18,19,20]. In addition to behaving as a prognostic biomarker, LIPI has been proposed as a method to classify patients subgroups with disparities in ICI treatment efficacy (i.e., a predictive biomarker) [13], albeit findings have been conflicting [8,14,15,21,22]. To date, there is minimal information of the prognostic performance of LIPI in patients with NSCLC initiating first-line, combination ICI approaches [11,12], and there is no information on treatment efficacy across LIPI defined subgroups in this cohort.

In post-hoc analysis of large randomized clinical trial data, this study aimed to evaluate the prognostic and treatment benefit predictive performance of LIPI in patients with NSCLC initiating first-line atezolizumab combination therapy.

## 2. Materials and Methods

### 2.1. Population

De-identified individual-participant data from clinical trial IMpower150 (ClinicalTrials.gov Identifier: NCT02366143, 15 September 2017 data cut-off) was utilized in this post hoc analysis (24, 25). IMpower150 involved patients with chemotherapy-naïve, metastatic non-squamous NSCLC randomized 1:1:1 to atezolizumab (1200 mg IV every 3 weeks) plus carboplatin (area under the concentration−time curve of 6 mg/mL/min for four or six cycles) plus paclitaxel (200 mg/m² IV every 3 weeks for four or six cycles (175 mg/m² IV for Asian patients (ACP)) or bevacizumab (15 mg/kg IV every 3 weeks), plus carboplatin plus paclitaxel (BCP), or atezolizumab plus BCP (ABCP) [23,24]. Conducted by F. Hoffmann-La Roche Ltd (Basel, Switzerland), IMpower150 was International Conference on Harmonization Good Clinical Practice guideline and the Declaration of Helsinki compliant, and participants provided written informed consent [23,24]. The post hoc analysis of di-identified data reported here was deemed negligible risk research and exempt from review by the Southern Adelaide Clinical Human Research Ethics Committee.

### 2.2. Predictor and Outcomes

The primary assessed outcome was overall survival (OS), with progression-free survival (PFS) assessed as a secondary outcome. In IMpower150, PFS was assessed by the investigator according to Response Evaluation Criteria in Solid Tumors (RECIST) version 1.1 [23,24].

The primary assessed covariate was the Lung Immune Prognostic Index (LIPI) designated upon a count of pre-treatment lactate dehydrogenase (LDH) >upper limit of normal (ULN) and derived neutrophil to lymphocyte ratio (dNLR calculated as neutrophil count/(white blood cell count–neutrophil count)) >3; good LIPI (0 factors), intermediate LIPI (1 factor), and poor LIPI (2 factors).

Pre-treatment age, sex, Eastern Cooperative Oncology Group performance status (ECOG PS), race, smoking status, histological subtype, effector T-cell gene signature score (Teff) [23,24], PD-L1 expression [23,24], epidermal growth factor receptor (EGFR) mutation status, and presence of liver metastases data were available.

### 2.3. Statistical Analysis

Cox proportional hazards regression, reported as hazard ratios (HR) with 95% confidence intervals (95% CI), was utilized to assess the prognostic significance of LIPI with survival outcomes in IMpower150 participants who were assigned an anti-cancer treatment containing atezolizumab. All statistical regressions were stratified by sex, PD-L1 expression, and presence of liver metastases (the randomization factors used in IMpower150 [23,24]. Complete case analyses adjusted for age, ECOG PS, race, smoking status, histology, Teff score, and EGFR mutation status were conducted. Adjustment variables were selected based upon biological plausibility, with the analyses undertaken to understand the independence of LIPI from other commonly prognostic variables. Discrimination performance of LIPI was measured using the concordance (c) statistic.

The statistical interaction between LIPI and ABCP/ACP for OS and PFS effect compared to BCP was evaluated using the intent-to-treat (ITT) population of IMpower150 [23,24,25]. Treatment effect heterogeneity was assessed using a treatment-by-LIPI interaction term within Cox proportional hazards regression. Testing of the proportional hazard assumption was conducted [26].

Statistical significance was set at a *p* value less than 0.05. Survival probabilities were estimated via Kaplan–Meier analysis. All analyses utilized R version 3.6.2 (R Core Team, Vienna, Austria).

## 3. Results

### 3.1. Population

Of 1148 (96% of 1202) IMpower150 participants with complete pre-treatment WBC count, neutrophil count and LDH data, 548 (48%) had good LIPI, 479 (42%) had intermediate LIPI, and 121 (11%) had poor LIPI (Appendix A). Median (95%CI) follow-up was 15 (15–16) months within the evaluable cohort. The OS and PFS effect (HR 95%CI) of ABCP (*n* = 385) versus BCP (*n* = 381) within the evaluable cohort was 0.78 (0.63–0.97) and 0.62 (0.53–0.74), respectively. The OS and PFS effect (HR 95%CI) of ACP (*n* = 382) versus BCP within the evaluable cohort was 0.85 (0.68–1.05) and 0.95 (0.81–1.12), respectively. Higher ECOG PS and liver metastases were associated with the poor LIPI group (Appendix A).

### 3.2. Prognostic Association

Within the 385 evaluable participants randomized ABCP, median (95%CI) OS ranged from 7 (4–13) months for the poor LIPI group to 24 (24-NA) months for the good LIPI group (*p* < 0.001, Table 1, Figure 1A). LIPI discrimination performance (c-statistic) was 0.63 for OS in the cohort. Median (95%CI) PFS ranged from 4 (3–7) months for the poor LIPI group to 12 (10–13) months for the good LIPI group (*p* < 0.001, c = 0.59, Table 1, Figure 1B). The LIPI OS and PFS association was demonstrated independent of age, sex, ECOG PS, race, smoking status, histological subtype, Teff, PD-L1 expression, EGFR mutation status, and presence of liver metastases within the ABCP treated cohort (Appendix A).

Within the 382 evaluable participants assigned ACP, median (95%CI) OS ranged from 7 (5–14) months for the poor LIPI group to 21 (20-NA) months for the good LIPI group (*p* < 0.001, c = 0.63, Table 1, Figure 1C). Median (95%CI) PFS ranged from 3 (2–6) months for the poor LIPI group to 7 (7–8) months for the good LIPI group (*p* < 0.001, c = 0.58, Table 1, Figure 1D).

### 3.3. Treatment Benefit Differences by LIPI Group

In the ITT population of IMpower150, the estimated absolute benefit in median OS for ABCP (compared to BCP) range from 1 month for the good LIPI group to no observed benefit in the poor LIPI group (Figure 2A–C). The relative OS benefit (HR 95%CI) of ABCP versus BCP was 0.78 (0.53–1.15), 0.67 (0.49–0.91), and 0.87 (0.51–1.47) for the good, intermediate, and poor LIPI groups, respectively (P(interaction) = 0.66, Table 2), which indicates a trend towards decreased OS benefit in the poor LIPI group did not constitute a statistically significant interaction. Assumptions of proportionality were upheld (*p* > 0.05) in assessment of OS within the good and intermediate LIPI groups. Within the poor LIPI subgroup non-proportionality was detected (*p* = 0.038).

The estimated absolute benefit in median PFS for ABCP (compared to BCP) range from 3 months for the good LIPI group to no observed benefit in the poor LIPI group (Figure 2D–F). The relative PFS benefit (HR 95%CI) of ABCP versus BCP was 0.59 (0.45–0.77), 0.52 (0.40–0.66), and 0.89 (0.56–1.42) for the good, intermediate and poor LIPI groups, respectively (P(interaction) = 0.13, Table 2). Assumptions of proportionality were upheld (*p* > 0.05) in an assessment of PFS outcomes.

Similar changes in the magnitude of OS and PFS effect were observed for ACP versus BCP, according to LIPI groups (Table 2, Appendix A).

## 4. Discussion

In a large cohort of participants with chemotherapy-naïve, metastatic non-squamous NSCLC we independently validated, for the first time, that LIPI is a prognostic marker that discriminates subgroups with distinctly different OS and PFS outcomes with atezolizumab combination therapies (ABCP or ACP). Although the median OS and PFS benefit of ABCP (compared to BCP) in the poor LIPI group was consistently less than the benefit observed in the intermediate and good LIPI groups, the interaction tests did not reach statistical significance. As such, at this stage, there is no conclusive evidence that LIPI can identify subgroups with impacted treatment benefit.

The study is strengthened by the sample size, high quality data. and random treatment allocation that enables valid assessments of treatment heterogeneity. To the best of the authors’ knowledge, this is the largest evaluation of the prognostic performance of LIPI in patients with NSCLC initiating first-line ICI combination approaches and the first to evaluate potential impacts on treatment benefit. In this study LIPI is confirmed a prognostic marker of OS and PFS in patients with chemotherapy-naïve, metastatic non-squamous NSCLC initiating ABCP or ACP therapy. The observed discrimination performance was consistent with prior reports (c-statistic range 0.59 to 0.63) [8,13,14], although the c-statistic is modest by prediction modelling standards [27,28]. Nonetheless, similar to prior studies, the spread of OS and PFS between the LIPI groups represent differences meaningful within the clinic and to patients, and thus there is significant potential for LIPI to support shared decision-making processes.

Compared to prior studies focused on patients with platinum-based chemotherapy resistant, advanced NSCLC [8,13,14], the present study included only chemotherapy-naïve, metastatic non-squamous NSCLC patients. The observed distribution of 48%, 42%, and 11% of participants within the good, intermediate, and poor LIPI groups, respectively, were similar to the original distributions of Mezquita et al [13] at 38%, 48%, and 15%, respectively. Not surprisingly, observed median OS and PFS according to LIPI groups were higher within our first-line cohort, than the prior pre-treated cohorts [8,13,14]. For example, median OS within our cohort treated with ABCP was 24, 16 and 7 months for the good, intermediate, and poor LIPI groups, respectively. Comparatively, median OS for the good, intermediate and poor LIPI groups were 17, 10, and 5 months, respectively, for Mezquita et al. [13], and 18, 11, and 5 months, respectively, for Sorich et al. [8].

A notable finding of the study was that median OS and PFS benefits of ABCP and ACP (compared to BCP) were consistently less in the poor LIPI group than in the intermediate and good LIPI groups. For example, the estimated absolute benefit in median PFS for ABCP (compared to BCP) was 3 months in the good LIPI group with no observed benefit in the poor LIPI group. The estimated decrease in OS and PFS benefit in the poor LIPI group did not constitute a statistically significant interaction, which is consistent with prior studies demonstrating that LIPI is also a prognostic marker for traditional chemotherapy options [8,14,15,21,22]. Nonetheless, it is recognized that clinical trials are not typically powered to detected treatment-by-covariate interactions. Thus, the consistent trend towards reduced benefit of ABCP within the poor LIPI group highlights a need for further research, particularly considering the poor LIPI group was only 11% of the sample and non-proportionality was detected within this group. A further limitation of this post hoc analysis is it solely focusses on atezolizumab combination approaches and it does not provide information on LIPI performance with alternate ICIs. It is also acknowledged that clinical trial inclusion criteria can limit the generalizability of findings. For example, IMpower150 only included participants eligible for bevacizumab, with no symptomatic or untreated brain metastases, no autoimmune diseases, and an ECOG PS of 0 or 1 [23,24]. Future research should therefore aim to (1) pool multiple clinical trials with longer follow up to boost the power to conclusively determine treatment effects within the poor LIPI group, (2) evaluate the performance of LIPI for other ICI combination approaches, and (3) evaluate LIPI prognostic performance within a large real-world cohort. Such research needs to be conducted in a timely manner, as while the potential of LIPI has now been demonstrated across several studies, it is acknowledged that the marker is not routinely used in clinical practice despite the increasing use of ICIs.

## 5. Conclusions

LIPI was validated as a significant and independent prognostic marker of OS and PFS in patients with chemotherapy-naïve, metastatic non-squamous NSCLC who initiated either ABCP or ACP therapy. With respect to treatment benefit, there was insufficient evidence to conclusively determine OS or PFS benefit of ABCP (compared to BCP) was reduced within the poor LIPI group, but it does warrant further investigation.

## Figures and Tables

**Figure 1 cancers-13-01176-f001:**
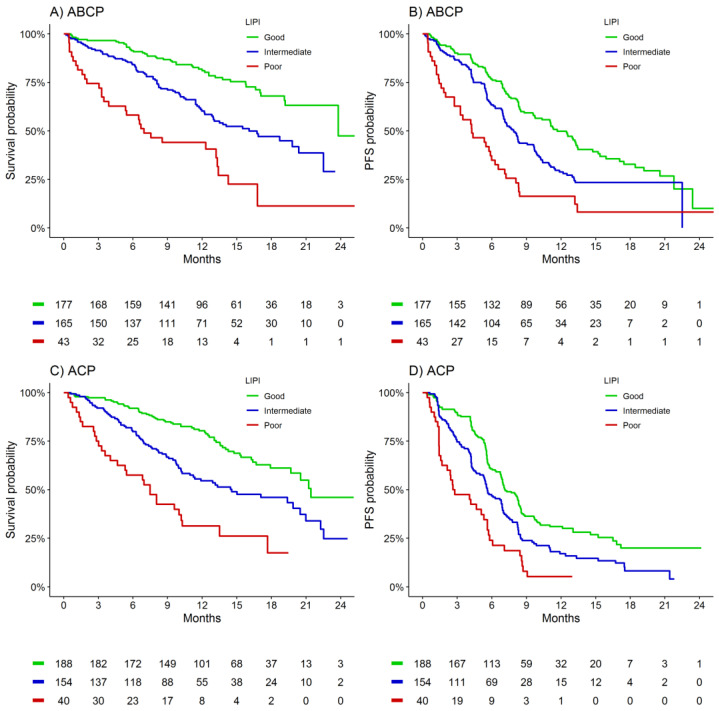
Kaplan-Meier estimates of overall survival and progression-free survival according to LIPI group for patients treated with ABCP or ACP. (**A**) ABCP—overall survival (**B**) ABCP—progression-free survival (**C**) ACP—overall survival (**D**) ABCP—progression-free survival.

**Figure 2 cancers-13-01176-f002:**
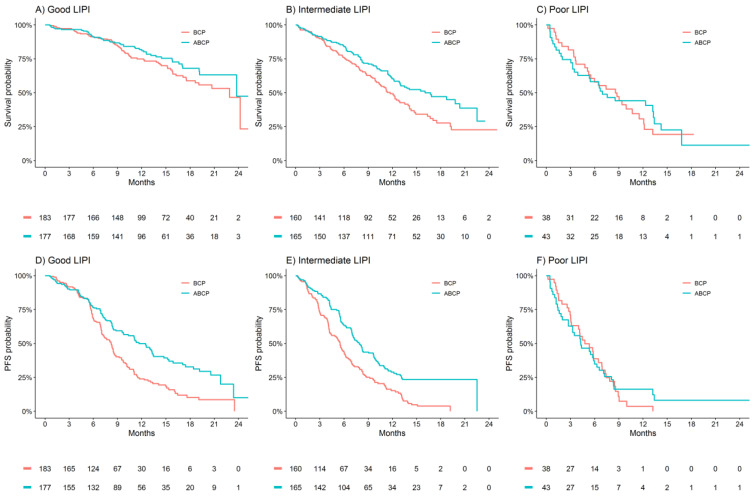
Kaplan–Meier estimates of overall survival and progression-free survival in the randomized arms of IMpower150 (ABCP versus BCP), subgrouped by LIPI. (**A**) Good LIPI—ABCP vs. BCP (**B**) Intermediate LIPI—ABCP vs. BCP (**C**) Poor LIPI—ABCP vs. BCP (**D**) Good LIPI—ACP vs. BCP (**E**) Intermediate LIPI—ACP vs. BCP (**F**) Poor LIPI—ACP vs. BCP.

**Table 1 cancers-13-01176-t001:** Association between lung immune prognostic index (LIPI) and survival outcomes for patients randomized atezolizumab-bevacizumab-carboplatin-paclitaxel (ABCP) or atezolizumab-carboplatin-paclitaxel (ACP).

Treatment Arm/Risk Group	OS				PFS			
	Median (95%CI) (Months)	HR (95%CI)	*p*	*c*	Median (95%CI) (Months)	HR (95%CI)	*p*	*c*
ABCP			<0.001	0.63			<0.001	0.59
Good LIPI (*n* = 177)	24 (24-NA)	1.00			12 (10–13)	1.00		
Intermediate LIPI (*n* = 165)	16 (13-NA)	2.16 (1.47–3.18)			8 (7–10)	1.47 (1.11–1.95)		
Poor LIPI (*n* = 43)	7 (4–13)	5.28 (3.20–8.69)			4 (3–7)	3.02 (2.03–4.50)		
ACP			<0.001	0.63			<0.001	0.58
Good LIPI (*n* = 188)	21 (20-NA)	1.00			7 (7–8)	1.00		
Intermediate LIPI (*n* = 154)	15 (11–21)	1.94 (1.37–2.76)			6 (5–7)	1.50 (1.17–1.93)		
Poor LIPI (*n* = 40)	7 (5–14)	5.05 (3.13–8.13)			3 (2–6)	2.86 (1.96–4.18)		

CI = confidence interval, HR = hazard ratio, PFS = progression free survival, OS = overall survival. Analyses stratified by sex, PD-L1 expression, and presence of liver metastases.

**Table 2 cancers-13-01176-t002:** Treatment effect modification by LIPI group.

Outcome	Good LIPI HR (95% CI)	Intermediate LIPI HR (95% CI)	Poor LIPI HR (95% CI)	P(interaction)
ABCP versus BCP				
	*n* = 360	*n* = 325	*n* = 81	
OS	0.78 (0.53–1.15)	0.67 (0.49–0.91)	0.87 (0.51–1.47)	0.66
PFS	0.59 (0.45–0.77)	0.52 (0.40–0.66)	0.89 (0.56–1.42)	0.13
ACP versus BCP				
	*n* = 371	*n* = 314	*n* = 78	
OS	0.90 (0.63–1.29)	0.76 (0.56–1.04)	1.08 (0.63–1.84)	0.51
PFS	0.96 (0.76–1.22)	0.89 (0.70–1.14)	1.45 (0.91–2.33)	0.19

CI = confidence interval, HR = hazard ratio, PFS = progression free survival, OS = overall survival. Analyses stratified by sex, PD-L1 expression, and presence of liver metastases.

## Data Availability

Data were accessed according to Roche’s policy and process for clinical study data sharing and is available for request at https://vivli.org/.

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
