# Peer review of "Value of the Lung Immune Prognostic Index in Patients with Non-Small Cell Lung Cancer Initiating First-Line Atezolizumab Combination Therapy: Subgroup Analysis of the IMPOWER150 Trial"

_cancers, 2021, doi:10.3390/cancers13051176_

Round 1

Reviewer 1 Report

Researchers have been looking for better prognostic or predictive markers in cancer immunotherapy. One of the candidates reported in the literature is the Lung Immune Prognostic Index (LIPI) according to LDH and derived neutrophil to lymphocyte ratio. Most previous reports are small scaled and/or in the second-line settings. The authors used publicly available patient-level data in one of the largest phase III trials (IMPower 150) that led to regulatory approval of atezolizumab, an anti-PDL1 antibody, in first line stage IV non-small cell lung cancer in the setting of chemo-immunotherapy.

Authors found that low LIPI score is associated with poor OS and PFS, and that it is not necessarily predicts survival benefit of Atezolizumab when added to backbone chemotherapy regimens.

My only comment is how authors can explain the benefit of atezolizumab in overall survival and PFS for the overall population that was reported in the original NEJM paper. For instance, in Figure 2, there is only 1 month or no improvement in OS in the 3 LIPI groups. 

Author Response

Response: We thank the reviewer for their time in reading and reviewing the manuscript, it is much appreciated. We have updated the manuscript to include a summary of the atezolizumab treatment effect in the overall evaluable population.

Changes: The results have been updated to include –

‘The OS and PFS effect (HR 95%CI) of ABCP (n=385) versus BCP (n=381) within the evaluable cohort was 0.78 (0.63-0.97) and 0.62 (0.53 -0.74), respectively. The OS and PFS effect (HR 95%CI) of ACP (n=382) versus BCP within the evaluable cohort was 0.85 (0.68-1.05) and 0.95 (0.81-1.12), respectively.’

Reviewer 2 Report

Value of the Lung Immune Prognostic Index in patients with non-small cell lung cancer initiating first-line atezolizumab combination therapy: subgroup analysis of the IMPOWER150 trial

Hopkins et al

This report describes an analysis of the lung immune prognostic index (LIPI) in the IMPOWER150 lung cancer clinical trial. It confirms/validates that LIPI is a prognostic marker in the 1st line stage IV non-small cell lung cancer setting, whilst being unable to confirm that LIPI is a predictive marker for treatment benefit. This is an important addition to our existing knowledge.

I thought the article was concise and its methods were sound. Results were presented iteratively and straightforward to follow for the general readership. Figures were clearly presented.

In order to maximise its potential, I would suggest the following changes:

  1. Corresponding table numbers to the text. 385 evaluable pts are described in 2nd results para but Table S1 suggests 400 pts for ABCP (presumably before losing 15 for evaluability). Similar for the two other treatment groups. Amend Table S1 with an extra row to highlight evaluable pt numbers.
  2. End of results para 2. I wasn’t clear on the process for how the model was confirmed for multivariate testing. Which of these prognostic variables were significant on univariate testing and what p value was used for their addition to the multivariate model? More detail would be helpful, potentially both in methods and results sections.

Author Response

Response: We thank the reviewer for their time to read and comment on the manuscript, it is much appreciated. We have updated Table S1 to indicate evaluable patients. Note that no multivariable models were built within the manuscript. Rather adjusted analyses, with variables selected based on biological plausibility, were undertaken with the intention to understand the independence of LIPI. The methods have been updated to indicate this difference.

Changes: Table S1 has been updated to indicate the number of evaluable patients per treatment.

The methods have been updated to include:

‘Adjustment variables were selected based upon biological plausibility, with the analyses undertaken to understand the independence of LIPI from other commonly prognostic variables. Discrimination performance of LIPI was measured using the concordance (c) statistic.’

Reviewer 3 Report

Please do implement the discussion section, highlighting the clinical importance of such a score in routine practice. Of note, the lack of LIPI score evaluation as a monitoring tool during the differing treatment courses should be mentioned as a limitation. 

Author Response

Response: Thank you to the reviewer for commenting on the manuscript. Like the reviewer we are excited about the potential clinical implications of LIPI and understand that current utilisation in practice is limited. We have expanded the discussion to highlight this point.

Changes: The discussion has been updated to include –

‘Such research needs to be conducted in a timely manner, as while the potential of LIPI has now been demonstrated across several studies, it is acknowledged that the marker is not routinely used in clinical practice despite the increasing use of ICIs.’

Reviewer 4 Report

The authors evaluated Lung Immune Prognostic Index in NSCLC patients.

Novel work. The authors state clearly the originality of their work in the discussion section.

Furthermore, some comments, maby in the introduction concerning LIPI and other  (lung) cancers is needed.

Author Response

Response: We appreciate the review and that the introduction should indicate that LIPI has shown prognostic performance for cancer types other than just lung cancers.

Changes: The introduction has been updated to include –

‘Studies have also demonstrated that LIPI may differentiate prognosis in patients treated with ICIs in other cancer types, including breast cancer, hepatocellular carcinoma, melanoma, renal cell carcinoma, small cell lung cancer, and urothelial carcinoma.’